# Microscale Alkenone Heterogeneity and Replicability of Ultra-High-Resolution Temperature Records from Marine Sediments

Jannis Viola<sup>1,2</sup>, Lars Wörmer<sup>2</sup>, Kai-Uwe Hinrichs<sup>2</sup>, Thomas Laepple<sup>1,2</sup>

<sup>1</sup>Alfred-Wegener-Institut, Helmholtz Center for Polar and Marine Research, Potsdam, D-14473, Germany <sup>2</sup>University of Bremen, MARUM – Center for Marine Environmental Sciences and Faculty of Geosciences, Bremen, D-28334, Germany

Correspondence to: Jannis Viola (jannis.viola@awi.de)

Abstract. The alkenone-derived  $U_{37}^{k'}$  proxy is crucial for the reconstruction of past sea surface temperatures in marine sedimentary archives. Recent advances in mass spectrometry imaging (MSI) now allow to measure alkenone abundance at the micrometer scale. Such an approach can theoretically provide proxy records as highly resolved as observational records and hold the promise of continuously reconstructing climate variability from subseasonal or interannual to centennial and millennial timescales. However, due to processes occurring during and after deposition, as well as during sampling and measurement, it is unclear how much climate signal is preserved in the proxy signal at these small spatial scales. Here, we investigated this question on sediment records from the Santa Barbara Basin (SBB), off California. We performed replicated MSI measurements on sediments with varying degrees of lamination to analyze the spatial structure and spatial reproducibility of the alkenone signal. We find that alkenone distributions are spatially heterogeneous even within laminae but exhibit small scale clustering over the range of ~0.5-1 mm. Measurements along laminated horizons show longer ranges of similarity and less overall variability compared to measurements across depth. Signal to noise ratios (SNR), the amount of shared variance between proxy records derived from the replicates across varying sediment conditions range from ~1 SNR at interannual resolution to ~3 SNR at subdecadal timescales.

MSI-based  $U_{37}^{K'}$  records in the SBB, supported by careful estimation of noise and uncertainty, thus can capture subdecadal SST variability and provide an upper limit for the signal content of Holocene and late Pleistocene SST reconstructions.

The approach presented here can be used in other settings to infer optimal sampling and measurement resolution, as well as to provide uncertainty estimations for the proxy records.

#### 1 Introduction

25

Understanding past climate and its dynamics is crucial for contextualizing recent climate change and projected future conditions. Sea surface temperature (SST) is an essential climate variable (Bojinski et al., 2014), but reliable SST observations only cover the last 150 years (Huang et al., 2017). Therefore, the instrumental record is too short to fully characterize climate variability at decadal or slower timescales (Ault et al., 2013; Laepple and Huybers, 2014). Similarly, to gain a better

35

understanding of the long-term dynamics of phenomena like monsoon or El Niño-Southern Oscillation (ENSO), records are required that can resolve interannual variability over longer time periods (Huang et al., 2017). Marine sediments offer such long, continuous archives and, in the case of laminated sediments, represent a key resource when aiming for the highest possible resolution (Hathorne et al., 2023; Schimmelmann et al., 2016). Recent technological advances in ultra-high resolution methods such as µXRF geochemical spatial scans (Blanchet et al., 2021), hyperspectral imaging (Butz et al., 2015; Zander et al., 2022) as well mass spectrometry imaging (MSI) (Alfken et al., 2020; Napier et al., 2022; Obreht et al., 2022; Wörmer et al., 2014, 2022) offer the potential to interrogate these archives with a spatial resolution leading to near-annual timescales while covering long enough intervals to capture the influence of slower variations of the climate system.

In traditional proxy studies, sampling resolution (mm to cm) typically exceeds archive accretion rates (de Winter et al., 2021).

40 This might, however not be the case for the microscale methods mentioned above, where theoretical resolution might exceed the actual temporal resolution at which proxies record climate signals. The temporal resolution and inter-record replicability of proxies can also be also affected by microscale heterogeneity in sediments or signal carriers. Such heterogeneity may arise during signal production, for example due to heterogeneous SST patterns, eddies, surface mixing, or variations in habitat depth. In marine settings, variability can also stem from differences in sinking rates, advection, and lateral transport. Post-depositional processes like bioturbation — even under low-oxygen conditions — can further alter the signal (Bernhard et al., 2003), introducing non-climatic variability similar to 'stratigraphic noise' (Fisher et al., 1985) in ice climate archvies, which reduces the reliability of single-proxy climate reconstructions (Münch et al., 2016). Bioturbation and sediment mixing can, for example, smooth temperature signals (Anderson, 2001; Hülse et al., 2022; Liu et al., 2021; Schiffelbein, 1984), leading to loss of resolution and dampening of high-frequency variability (Münch and Laepple, 2018). 2D methods that map the distribution of the signal carriers harbor the potential to estimate the spatial variability of the signal of the otherwise discretely sampled or line-scanned proxies, ultimately deriving estimates of the proportion of climate variability captured.

In this study, we focus on long-chain alkenones produced by haptophyte algae. The alkenone-based  $U_{37}^{K'}$  (Prahl et al., 1988; Prahl and Wakeham, 1987) is a well-established SST proxy with global spatial and long temporal coverage. Additionally, we investigate pyropheophorbide  $\alpha$  — a chlorophyll degradation product (Goericke et al., 2000) — as a first-order proxy for primary productivity. The sediment cores used are from the Santa Barbara Basin (SBB), offshore Southern California. Seasonal runoff, high primary productivity, and the basin's bathymetry allow for the formation of laminae or annual varve couplets, and for their preservation under low-oxygen conditions that reduce bioturbation (Schimmelmann and Lange, 1996; Thunell et al., 1995). Past variations in bottom-water oxygen levels resulted in varying bioturbation intensities, from minimal disturbance that preserved varves to complete mixing after colonization by larger fauna (Anderson et al., 1989). We present spatially resolved alkenone measurements at 100  $\mu$ m resolution to characterize the spatial heterogeneity of the biomarkers in relation to lamina preservation, from well-preserved to mixed intervals. Mixing and bioturbation intensities assessed in this study are representative for the Holocene and late Pleistocene SBB sediments (Anderson et al., 1990; Behl, 1995). We then assess the influence of microscale heterogeneity on noise level and replicability of MSI-derived high-resolution  $U_{37}^{K'}$  SST proxies.



Subsequently, we estimate signal-to-noise ratios (SNR) of individual MSI-based reconstructions which indicate the finest usable proxy time series resolution based on the sediment structure.

#### 2 Materials and Methods

# 2.1 Samples, Sample Processing and Measurements

Sediment core MV1012-001KC was retrieved by research vessel Melville during cruise MV1012 in 2012, stored and accessed at Scripps Institution of Oceanography's cored sediment and microfossil collection. We used a 30-cm long section of this core, named "KC1" in the following. Depths are expressed as relative cm starting at the top of the sampled section. We additionally utilized data of three 5 cm replicated MSI measurements (approx. years 1913-1935) of boxcore SPR0901-05BC originally published by Alfken et al. (2020) as an example for well-preserved varved deposition with known correlation to the instrumental record, named "SBX" in the remainder of the study. The sediment samples were treated following the procedure described by Alfken et al. (2019). In brief, after subsampling, X-ray photographs of the wet sediment slabs were taken using a Faxitron 43855A X-ray cabinet (Hewlett-Packard). 5-cm subsections were freeze dried and embeddedd in an aqueous gelatine:CMC (4%:1%) solution (Gelatine from porcine skin, gel strength 300, type A and sodium carboxymethylcellulose, Sigma-Aldrich Chemie GmbH, Munich Germany). After freezing of the embedded samples, they were cut into 100-µm slices using a cryomicrotome (Medite Cryostat M630) and the slices were placed on indium-tin-oxide coated (ITO) glass slides (Bruker Daltonik GmbH, Bremen, Germany). High-resolution RGB scans were performed as guidance for setting up the FT-ICR-MS measurements. Finally, the slices were measured using a 7T solariX XR FT-ICR-MS coupled to a LDI source equipped with a Smartbeam II laser (Nd:YAG, UV-A Laser, 355nm wavelength, Bruker Daltonik, Bremen). Details on FT-ICR-MS measurement settings can be found in the supplements section S1.

For each 5 cm depth interval of KC1 00-30 cm relative depth, the workflow was repeated on three subsequent slices, yielding triplicate measurements with the minimum achievable distance to the previous measurement area (100-µm cutting thickness) (see Fig. 1).

The MSI measurements were inspected and lockmass calibrated using Bruker Compass DataAnalysis 5.0 SR1 (Bruker Daltonik Gmbh, 2017). Our main targets were the Na<sup>+</sup> adducts of the di- and triunsaturated alkenones (C37:2, C37:3). Calculated  $U_{37}^{K'}$  values were translated to temperature using the calibration of Prahl and Wakeham (1987). Other calibrations are available (Tierney and Tingley, 2018), yet the experimental calibration of Prahl and Wakeham was confirmed by Müller et al. (1998) for 60S-60N and the applicability of this calibration for California specifically was supported by Herbert et al. (1998). As our results focus on replicability, they are not sensitive to the choice of calibration. Besides alkenones, we utilized pyropheophorbide  $\alpha$  as a first order estimation for primary production and calibrant during FT-ICR-MS lockmass calibration

due to its ubiquity in SBB sediments (Liu et al., 2022) and stability over the timescales of this study (Szymczak-Żyła et al., 2011).

Figure 1: Schematic representation of the triplicate workflow. 1) 30-cm core LL-channel subsampling (Suigetsu 2006 Project Members and Nakagawa, 2014), used for X-ray photography; 2) freeze drying and embedding of 5-cm samples; 3) cryomicrotome cutting of 100-µm slices to be placed on ITO slides, 4) MSI scans, 5) horizon-wise aggregation & conversion to time series, 6) time series analyses.

#### 2.2 Study Area




The Santa Barbara Basin is located in the Southern California Bight and part of the California Current System (Bograd et al., 2019). The basin covers ~110 km² and is separated from the Pacific in the west and the Santa Monica Basin in the east by submarine sills, with the depocenter at ~595m (Soutar and Crill, 1977). The bathymetric layout together with seasonal upwelling, high productivity in summer and high sedimentation rates from winter runoff can favor the preservation of laminated or even varved sediments (Reimers et al., 1990; Thunell et al., 1995). The sediment sequence is interrupted by flood layers ("gray layers") and mass movements from the upper shelf close to the shore ("massive olive layers") and bioturbated intervals (Du et al., 2018; Hendy et al., 2013). Oxic time intervals allow the colonization with macrofauna, for example during the Macoma event 1835-1840 (Schimmelmann et al., 1992). During the Holocene and Late Pleistocene, varved, laminated or bioturbated sections alternate in response to phases of varying oxygenation (Anderson et al., 1990). However, Bernhard et al. (Bernhard et al., 2003) report microscale bioturbation even in oxygen limited or anoxic-sulfidic conditions.

## 2.3 Age Control

For this study, we focused on a sediment section that includes both partially laminated and mixed intervals, beginning around the 1761 AD flood layer (Hendy et al., 2013). The presence of a massive olive-colored layer and partially mixed sections below prevented the development of a detailed age model beyond linear interpolation between the 1761 AD and 1532 AD gray flood layers, following approaches such as Zhao et al. (Zhao et al., 2000, p.200) and O'Mara et al. (O'Mara et al., 2019). As a result, we conducted our analyses on a depth scale, which also enhances the comparability of our results across different depths and sediment structures within the Santa Barbara Basin.






# 2.4 Spatial Analyses

We estimated the spatial correlation structure of the measured compounds and spotwise-calculated  $U_{37}^{K'}$  ("swUk") temperatures using variogram analysis (Cressie, 1993; Maroufpoor et al., 2020; Wikle et al., 2019). We compared differences in horizontal (within horizon) and vertical (downcore) spatial (dis)similarity to quantify the zonal and geometric anisotropy, to assess the influence of lamina preservation and the homogeneity of deposited horizons. Intensities of alkenones and pyropheophorbide  $\alpha$  were log10 transformed, scaled and centered to account for strong nonnormality of the data, while spotwise  $U_{37}^{K'}$  (swUk) values were used without transformation. For the estimation of the sample variogram we used a gstat implementation (Gräler et al., 2016; Pebesma, 2004), choosing 100  $\mu$ m bin widths, together with horizontal and vertical tolerance of 10° to ensure sufficient data coverage and account for deviations in sample or laminae orientation.

# 2.5 Time-series Analyses

MSI data was mapped to the X-ray density maps using three tie points and an affine transformation to correct for potential sample deformation during embedding and cutting (Alfken et al., 2020) using the python software msiAlign (Liu et al., 2025). The known flood layer in the upper part of the KC1 segment was identified using X-ray density scans and MSI data, and the respective subsection was excluded from the 0-5 cm interval replicates prior to statistical analyses. Horizons with very low MSI detections across all compounds (fewer than 10 simultaneous detections of C37:2 and C37:3) were also removed. To avoid introducing gaps and to ensure sufficient detections per horizon, the minimum horizon width was set to 200 µm. Short segments of missing horizons due to insufficient detections were linearly interpolated. On average, ~4% of horizons were missing: ~1.4% at series ends and ~2.7% within.

Following Laepple and Huybers (2014), power spectral density (PSD) estimates were calculated on linearly detrended, centered data using the multitaper method with three tapers and a time-bandwidth parameter  $\omega = 2$  (Thomson, 1982). The PSD estimates were smoothed using a Gaussian kernel with a constant width of 0.2 on the logarithmic timescale (base 10) (Kirchner, 2005) and the three lowest and highest frequencies were omitted.

We followed the framework developed by Münch and Laepple (2018) based on a partitioning of variance approach (Fisher et al., 1985) to separate the climatic signal and noise contributions in the spectral domain. We defined the climate signal as the common signal shared between the time series of a depth interval. The residual, individual variations formed the noise component and allowed the quantification of the shared signal as signal-to-noise ratios.

Given *n* sediment slices that share a common signal, the mean power spectrum, *M*, computed by averaging the individual spectra of all slices, provides a precise estimate of the underlying can correspond to proxy spectrum *P*. In contrast, the power spectrum, *S*, of the stacked record from averaging all datasets in the time domain, will also contain the full common signal, but with the noise proportions reduced by a factor of *n*. By combining both quantities one can derive expressions for the climate

and noise spectra (McPartland et al., 2024; Münch and Laepple, 155 2018)

$$C = \frac{n}{n-1} (S - \mathcal{M}/n) \tag{1}$$

$$\mathcal{N} = \frac{n}{n-1} \left( \mathcal{M} - \mathcal{S} - \frac{n-1}{n} \right) \tag{2}$$

with the ratio of *C:N* yielding the frequency-resolved signal-to-noise ratio (SNR). We further compute the integrated SNR as equals the SNR of the time-series that one obtains if one measures at some specific resolution

$$iSNR_{f(Nyq)} = \frac{\int_0^{fNyq} c_f}{\int_0^{fNyq} N_f} \tag{3}$$

#### 3 Results



#### 3.1 Characterization of MSI maps

The measurement areas ranged from 14907 to 26937 laser shots for each 5-cm piece (20957 +- 4134 =  $1\sigma$ ), and thus ~40 shots per 100- $\mu$ m horizon. On average, at 62% of the shots at least one of the three compounds was detected. Pyropheophorbide  $\alpha$  was detected in 61% of the spectra, while at least one alkenone was detected in 46% and both in 34%. When aggregating 200- $\mu$ m horizons to time series, an average of ~37 shots per horizon yielded at least one alkenone compound, and ~31 yielded both (Table 1)

We first investigate the replicability and signal content of individual MSI maps in terms of the similarity of their swUk (spotwise-calculated  $U_{37}^{K'}$ ) values and distributions. The maps of swUk values show strong spatial heterogeneity (Figure 2A). Despite this, the average values were highly consistent between MSI replicates: the standard error of averaging mean swUks per triplicate was just 0.001, corresponding to ~0.03°C (Prahl et al., 1987). Similarly, the level of spatial variability was consistent across replicates. The average swUk variance within maps ('field variance') was 0.00595 ± 0.00004, corresponding to 2.34 ± 0.19 °C. The swUk distributions were similar across depth intervals and unaffected by sediment structure (see Fig. 2C).

Values from the same horizon can be aggregated into a single data-point, and thereby a time series can be constructed. The average variance of swUK (spotwise  $U_{37}^{K'}$ ) in these 200 µm horizons was similar to the observed total field variances:  $0.006 \pm 0.00005$  (2.35 ± 0.21°C). Visual comparison between replicate time-series shows good agreement in terms of mean values and slow variation (see Fig. 2, panel D) yet weak correlation of the fastest resolution of 200-µm steps, with an average Pearson's r of 0.12 and average RMSE of 0.03 (~0.89°C). Such weak correlation is most likely due to a lack of fine-scale depth alignment.

Figure 2: Maps and time-series of the sediment section KC1. (A) X-ray density map, (B) exemplary swUk (spotwise  $U_{37}^{K}$ ) maps of one replicate per depth interval. Maps are shown as measured, on the MSI coordinate grid, before affine transformation onto Xray coordinates, values below 1% and above 99% quantiles were removed for optimal color scaling during plotting. (C) Scaled density plots of the spatial swUk maps per replicate, and (D) the resulting  $U_{37}^{K'}$  time-series per replicate and depth interval. Note that the area corresponding to the 1761 AD floodlayer in depth interval 0-5 cm was removed prior to analyses.


#### 3.2 Microscale Analyses of Spatial Heterogeneity

After confirming consistency between replicates on the coarser, whole-slice level, we examined the spatial proxy heterogeneity using maps of spotwise-calculated Uk'<sub>37</sub> values (swUk). These spatial patterns show high variability both across depth and laterally, with average swUk variance across depths and horizons both approximating 0.00596 (vertical: 0.00597; horizontal: 0.00596).

To explore the structure of this heterogeneity in more detail, we focus on two representative depths—one laminated and one mixed—along with their respective replicates (Fig. 2). Notably, clear laminations seen in the X-ray scans (e.g., at 5–7 cm; Fig. 2A, top row) are not reflected in the swUk patterns. Instead, both laminated and mixed intervals show swUk values forming small clusters of neighboring pixels with similar temperatures.

Figure 3: Overview maps of the measurement areas for three replicates from 0-5 cm (A-C) and 10 -15 cm (G-I) depth: gray areas indicate spots in which alkenones were successfully detected. Pink line: onset of the 1761AD flood layer; black rectangles indicate


5x5mm inlets as shown in D-F and J-L. These zoom-ins show values for swUk (spotwise  $U_{37}^{K'}$ ) converted to SST (Prahl and Wakeham, 1987). D-F represent a laminated interval below the flood layer, while J-L correspond to a thoroughly mixed interval of KC1B ("olive turbidite", event 1D in Hendy et al. (2013)). Maps are shown as measured, before affine transformation onto Xray coordinates.

Table 1: Map statistics averaged across replicates per depth interval. For "swUk mean" spotwise-calculated  $U_{37}^{K'}$  values have been averaged for the replicate maps and subsequently per 5-cm depth interval. Accordingly, swUk variance is the average variance of swUk values ("field variance") of the replicate maps per depth interval.  $U_{37}^{K'}$  units have been rounded to three digits, S.E. below 0.001 (~0.03°C) are not shown.

| Depth | swUk          | swUk     | Number of    | Coverage [%]   | Coverage [%]     | Coverage [%]   |
|-------|---------------|----------|--------------|----------------|------------------|----------------|
| (cm)  | mean          | variance | spots        | (any compound) | (both alkenones) | (any alkenone) |
| 0-5   | 0.523 ± 0.001 | 0.006    | 14907 ± 358  | 49.3 ± 1.1     | 31.5 ± 0.7       | 39.8 ± 0.9     |
| 5-10  | 0.529         | 0.006    | 26937 ± 563  | 62.3 ± 0.7     | 38.8 ± 0.5       | 49.6 ± 0.6     |
| 10-15 | 0.538         | 0.006    | 20002 ± 353  | 35.4 ± 0.9     | 20.5 ± 0.8       | 27.6 ± 0.9     |
| 15-20 | 0.534 ± 0.001 | 0.006    | 22043 ± 1788 | 68.4 ± 2.1     | 29 ± 2.9         | 43.5 ± 2.6     |
| 20-25 | 0.534 ± 0.001 | 0.006    | 18582 ±      | 79.2 ± 4.5     | 42.9 ± 2.5       | 59.5 ± 3.4     |
| 25-30 | 0.528 ± 0.001 | 0.006    | 23269 ± 559  | 79.6 ± 1.2     | 40.1 ± 2.9       | 54.9 ± 2.5     |

In order to more precisely describe local (dis)similarities in biomarker patterns, we obtained averaged variogram estimates of swUk. This approach reveals that only a small portion of total variability is spatially structured (Table 2) in KC1. The total map-level variance, the "sill" in the variograms, is 0.00595 (~2.34 °C), of which just 0.0002 (~0.43 °C) shows small-scale spatial structure — i.e., clustering or increased similarity among neighboring values. Most of the variability (~93-99%, avg. 96%) is unstructured or unresolved, termed the nugget effect (Co). High contribution of C0 can point to variation which is either spatially unresolved or intrinsic variance of the proxy, or measurement noise. In contrast, pyropheophorbide  $\alpha$  shows a lower nugget component (80-96%, avg. ~86%), indicating stronger spatial structure and a slower decay of similarity with distance (Fig. 4A).


We then separated vertical from horizontal similarity to distinguish between variability across successive depositional layers (vertical) and within the same horizon (horizontal). As a proof of concept, we also applied this approach to a varved interval from a box core as a best-case reference without sediment mixing (Alfken et al., 2020, 2021). In both sample sets, this separation (solid lines: vertical; dashed: horizontal, Figs. 4) reveals geometric anisotropy: similarity decreases more slowly in horizontal than vertical directions. This can be interpreted as within-horizon variance (dashed) being lower and increasing more gradually with distance than between layers (solid). This pattern is most clearly visible in the abundance data of pyrophophorbide  $\alpha$ , but also still detectable in the C37:3 alkenone and  $U_{37}^{K'}$  data. As expected, differences are clearer in the varved section of SBX than in the combined core KC1. Notably, in the varved section pyrophophorbide  $\alpha$  and, to a lesser extent,  $U_{37}^{K'}$  show a sinusoidal pattern in similarity with peaks and troughs aligning with the annual sedimentation rate of ~1.4 mm (Thunell, 1998), variograms of individual sediment pieces can be found in the supplements (S4).

Figure 4: Variogram estimates of pyropheophorbide  $\alpha$  and C37:3 intensities and swUk (spotwise  $U_{37}^{K}$ ) temperatures, showing the average variability of pairwise spot comparisons ( $\gamma$ ) as a function of distance. A and B are obtained for the exemplary varved section of SBX, averaged over 3 replicates. C and D for sediment segment KC1, averaged over all replicates and depth intervals (3x 6x 5cm).

Note that each panel has individual y-axes, as pyropheophorbide  $\alpha$  and C37:3 values in A, C were scaled prior to calculation and are not in their native units, whereas in B, D  $\gamma$  values correspond to  $U_{37}^{K'}$  units.

Table 2: Estimated swUk variogram parameters averaged per piece.

| Depth    | Sill (total field | Variance without spatial | Variance with spatial | C0 contribution to total |
|----------|-------------------|--------------------------|-----------------------|--------------------------|
| [rel.cm] | variance)         | structuring (C0)         | structuring           | variance [%]             |
| 0-5      | 0.005871          | 0.005467                 | 0.000404              | 93.12                    |
| 5-10     | 0.005878          | 0.005595                 | 0.000284              | 95.19                    |
| 10-15    | 0.005946          | 0.005883                 | 6.32E-05              | 98.94                    |
| 15-20    | 0.006121          | 0.006038                 | 8.32E-05              | 98.62                    |
| 20-25    | 0.006094          | 0.005934                 | 0.00016               | 97.39                    |
| 25-30    | 0.00582           | 0.005612                 | 0.000208              | 96.42                    |

#### 3.3 Frequency-dependent Signal and Noise Components of Replicated Time-Series

- Replication within 5-cm depth intervals allowed us to quantify the reproducibility between individual MSI slices and estimate SNR of the resulting time series. The frequency dependent separation of the shared signal and independent noise shows that the replicates contain increasing signal content with timescale and flat noise spectra. Signal spectra showed a slope of  $\sim$ -1.7 on average estimated between f=1/10 mm<sup>-1</sup> and f=1/2 mm<sup>-1</sup>, the interval of the spectral estimates not affected by bias and gaps. The noise components showed no frequency dependency ( $\sim$ -0.13 between f=1/10 mm<sup>-1</sup> and f=1/2 mm<sup>-1</sup>, see Fig. 5A).
- As a reference, we added noise spectra derived from total field variance (black dotted line, Fig. 5A), downscaled by a factor of  $\sim$ 37 the average number of swUk measurements per horizon at 200  $\mu$ m resolution (see also Table 2) to illustrate how independent (measurement) noise scales with averaging. The actual noise level was  $\sim$ 13 higher than this baseline, indicating noise reduction weaker than 1/n when aggregating to horizonwise  $U_{37}^{K'}$  data.
- Integrating the spectra from high to low frequencies yields integrated SNR (*iSNR*), which is an estimate of the expected SNR at a given sampling resolution. *iSNR* values reach from ~0.24-~3.1, considering sampling from submillimeter to centimeter steps (Fig. 5B).

Figure 5: Frequency dependent Signal and noise components in comparison to field variance (A) and the resulting integrated SNR (iSNR) estimate (B). Dashed line shows the average variance of the swUk field in black; the dotted line is the field variance downscaled by the average number of swUk shots per horizon, imitating the noise reduction during time series processing.

We then assessed the influence of sediment structure on time domain signal-to-noise variance ratios (Fisher et al., 1985) (i.e. SNR between variance of the noise and signal componts) of the different depth intervals in comparison to the "best case" varved 5 cm section published earlier (Alfken et al., 2020, 2021). We also repeated the analysis using different aggregation steps (Fig. 6) to assess the impact of data processing choices. The tested sampling widths ranged from 200 to 1400 μm, which, under ideal preservation and an average sedimentation rate of 1.4 mm/yr (Thunell, 1998), would correspond to monthly to annual resolution. For core segment KC1 these SNRs ranged from ~0 in the most strongly mixed interval to ~2 in an interval with partially preserved varves (Fig. 6). In comparison, the varved section of SBX reached SNRs of ~2.8 on average.

Figure 6: Comparison of sediment structure and SNR based on time domain signal-to-noise variance ratios (Fisher et al., 1985), resulting from aggregating horizons from 0.2 to 1.4mm, the theoretical maximum aggregation for annual resolution (Thunell et al 1995). A and C show a 5 cm exemplary interval from the varved upper part of SBB (ca. 1913-1935). C and D show the KC1 segment which represents the different conditions found at SBB throughout the late Holocene, ranging from laminated sediments, varying degrees of mixing to turbidites. Note that the 1761AD flood layer in KC1 0 – 5 cm was removed from the timeseries prior to SNR estimation. C and D are X-ray density maps of the respective samples, brighter colors indicate denser material. The black holes are markers for orientation during sampling and to correct for distortion. Note that an exposure artefact was removed from the X-ray photography of KC1 at ~20-25 cm. An unaltered version can be found in Supplementary Fig. S3.

#### 4 Discussion





This study investigated the spatial heterogeneity of biomarkers at submillimeter resolutions and its effect on the time-scale-dependent signal-to-noise ratios of MSI-based high-resolution SST reconstructions. In the following sections, we discuss the observed spatial heterogeneity of biomarkers and the influence of sediment mixing intensity on the interpretability of high-resolution SST reconstructions.

#### 4.1. Spatial Heterogeneity of Biomarkers

Variogram analyses show greater similarity across sedimentary horizons than downcore (see Fig. 4), consistent with the expectation of layered archives where variation over depth includes additional variability from changing climate signals over time (Fisher et al., 1985; Münch and Laepple, 2018).

Biomarker maps reveal sub-millimeter-scale clustering and marked differences in spatial heterogeneity between compounds (Fig. 4, 5). For swUk, the variance showing spatial structuring is minimal (~4%). Under favorable sediment preservation






conditions, varved sections manifest as oscillations in the variogram, with wavelengths approximating the average sedimentation rate (Thunell, 1998).

The high proportion of unstructured variance in our biomarker and proxy maps suggests significant noise at the individual spot level. Aggregating data across horizons enhances the signal content because noise is largely uncorrelated between spots at the microscale. However, the study's data does not allow us to determine whether this noise stems from measurement errors or from the deposition of heterogeneous biomarker signals. On one hand, strong shot-to-shot variation is a common issue in MSI (Tobias and Hummon, 2020). However, classic bulk sediment  $U_{37}^{K'}$  estimates reflect averaged or smoothed signals, and it is difficult to estimate the heterogeneity they are capturing. For example, water column and sediment trap data display a broader range of  $U_{37}^{K'}$  values around the measured ocean temperatures than core tops, and lab cultures have been shown to exhibit even greater variability in biomarker ratios than observations in natural settings (Conte et al., 1995; Conte and Eglinton, 1993; Freeman and Wakeham, 1992; Herbert et al., 1998; Sikes and Volkman, 1993). Our observed proxy spot-level variance is smaller than that of the monthly SST estimates for SBB yet larger than the interannual variation (see supp. S5), suggesting that the observed heterogeneity may indeed originate from SST and capture variability typically lost in bulk sediment samples. Therefore, MSI-based  $U_{37}^{K'}$  maps can help reconcile steps along the proxy formation chain.

The small-scale clustering (~0.5 mm) observed in the biomarker maps imply that, for optimal sampling, MSI measurement windows should be in the range of a few millimeters to be wide enough to 1) have enough successful shots to overcome sparse detections, and 2) avoid bias or correlated noise that could arise from only targeting single proxy clusters. Therefore, the typical measurement window widths of ~4-8 mm used in this and earlier studies (Alfken et al., 2020, 2021; Liu et al., 2022; Napier et al., 2022; Obreht et al., 2022; Wörmer et al., 2014, 2022) should effectively reduce the noise in downcore records.

# 4.2. Timescale dependent signal to noise estimations

Averaging across a diverse range of sediment preservation conditions in core segment KC1, we detect iSNRs around ~1 at interannual resolution, increasing to 3 at subdecadal (ENSO) resolution. The extracted noise component shows no dependence on time scale, while the variance of the signal components increases with time scale. This suggests that temporal aggregation effectively reduces noise in individual time series and enhances the relative potential climate signal content. Additionally, stacking multiple replicates would further decrease the noise contribution by a factor of 1/N, boosting the signal content.

The iSNR >> 1 indicates that individual MSI series are representative of their core samples and demonstrate potential for high-resolution climate reconstructions, at subdecadal resolution in the case of the SBB sedimentary archive containing laminated and mixed intervals. This aligns with findings from O'Mara et al. (2019), who estimated the effective temporal resolution of neighboring laminated SST records from San Lazaro Basin to approximately decadal without replication or stacking.

One limitation of our method is its sensitivity to depth misalignment between replicates, which could lead to a bias in SNR estimates at the fastest temporal and corresponding spatial scales («1mm). To minimize these effects, we used sediment marker holes and the mapping of MSI data onto X-ray imagery. More advanced image correction and alignment might further reduce






these biases in the future. However, low SNRs at 

temporal scale, reaching values suitable for subdecadal interpretations even when using single MSI measurements. In contrast, mixed sediments show reduced signal content, particularly at higher frequencies, limiting the resolution of interpretable variability.

To fully exploit the potential of MSI-based time series, further refinement of data processing methods is needed to minimize noise introduced during the conversion of spatial measurements to SST estimates. Regional replication is also essential to isolate shared climate signals and suppress locally correlated noise. With optimized workflows and stacking approaches, MSI-derived SST records from Santa Barbara Basin show strong potential to resolve interannual to centennial variability throughout the Holocene and Late Pleistocene, offering a powerful complement to existing paleoclimate archives.

# 370

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

#### 555 Code and Data Availability

The dataset has been submitted to Pangaea and will be available under DOI \_\_\_\_\_\_, the code used to process analyse the files and will be made available on zenodo (https://doi.org/10.5281/zenodo.17352962).

# Acknowledgements

We thank the scientists, technicians and support staff of cruises MV1012 and SPR0901 for the retrieval of the core material and especially Richard Norris and Janina Groninga for sampling the cores at Scripps Institution of Oceanography's geological collection. Furthermore we would like to thank Jenny Altun and Susanne Alfken for lab training and help during the measurements, as well as Weimin Liu for invaluable help during processing. LLMs have been used to improve flow of text, as well as to correct grammar and spelling.

Financial support

565

570

575

580

This project has received funding from the European Research Council (ERC) under the European Union's Horizon 2020 research and innovation programme (grant agreement no. 716092). Additionally, this research was supported by Germany's Excellence Strategy (EXC-2077) project 390741603 'The Ocean Floor – Earth's Uncharted Interface'.

590

585