# Peer review of "Microscale Alkenone Heterogeneity and Replicability of Ultra-High-Resolution Temperature Records from Marine Sediments"

_EGUsphere, 2025_

## Referee Comment (RC1)

**Review of Viola *et al.* for *Climate of the Past***

**Joseph B. Novak**

**RECOMMENDATION**

Minor revision.

**SUMMARY**

Viola et al. present a very nice study that explores the potential of mass spectrometry imaging to capture interannual climate variability in sedimentary archives. The data presented here probe the extent to which sedimentary processes may obscure these high-frequency climate signals. This study is an important contribution to the cutting edge of our field – particularly because of the profoundly important climate processes the MSI technique could be used to investigate. The writing and figure design are excellent.

My main comment relates to the mass spectrometry techniques used. I am not familiar with the statistical techniques used in this manuscript and leave their evaluation to the other reviewers.

I look forward to the publication of this work after my comments are addressed.

**MAJOR COMMENTS**

*Methods*:

Is there any reason for concern about potential differences in the ionization efficiency of the $C_{37:3}$ vs. $C_{37:2}$ alkenones influencing your results? This is an issue for other mass spectrometer techniques (e.g., GC-MS and HPLC-MS, see (Chaler et al., 2000, 2003; Liao et al., 2023)). If steps were taken to account for this on the analytical side, some additional text outlining that procedure would be helpful to better understand the data.

**MINOR COMMENTS**

**L29–32:** Another important aspect of paleoclimate archives is that they allow climate scientists to understand climate processes in warmer-than-present climate states. It may be worthwhile to add a comment to this effect.

**L40:** a comma is needed after "however."

**L42:** remove space between "carries" and the period at the end of the sentence.

**L55:** "export productivity" is a more appropriate term here than "primary productivity" because the sediments only preserve the proportion of the biological products that are exported from the surface and buried.

**L150–151:** I think there is a typo or missing word here.

**Figure 2 caption:** an explanation of the black "Xs" in panel A would be nice.

**L220:** need a subscript on "C0"

**L230:** $C_{37:3}$ needs a subscript

**REFERENCES**

Chaler, R., Grimalt, J. O., Pelejero, C., & Calvo, E. (2000). Sensitivity Effects in $U$ k ' 37 Paleotemperature Estimation by Chemical Ionization Mass Spectrometry. *Analytical Chemistry*, *72*(24), 5892–5897. https://doi.org/10.1021/ac001014q

Chaler, R., Villanueva, J., & Grimalt, J. O. (2003). Non-linear effects in the determination of paleotemperature Uk'37 alkenone ratios by chemical ionization mass spectrometry. *Journal of Chromatography A*, *1012*(1), 87–93. https://doi.org/10.1016/S0021-9673(03)01188-9

Liao, S., Liu, X.-L., Manz, K. E., Pennell, K. D., Novak, J., Santos, E., & Huang, Y. (2023). Comprehensive analysis of alkenones by reversed-phase HPLC-MS with unprecedented selectivity, linearity and sensitivity. *Talanta*, 124653. https://doi.org/10.1016/j.talanta.2023.124653

---

## Referee Comment (RC2)

**Review of Viola et al. 2025**

This study offers a valuable sensitivity test for a promising new tool in alkenone paleothermometry, mass spectrometry imaging (MSI), which currently offers the highest possible resolution for $U^{k'}_{37}$ measurements in sediment. Here, MSI is applied to a well-characterized, finely-laminated sediment core from the Santa Barbara Basin (SBB). The authors describe heterogeneities in $U^{k'}_{37}$ measurements across horizontal laminae and vertical depth, using estimates of SNR to infer the upper resolution limit at which a climate signal is preserved. The use of SBB sediments offers an opportunity for excellent age constraints and the description of signal preservation in temporal terms. This approach has a number of valuable applications in alkenone paleothermometry and is helpful for those looking to apply MSI in a statistically rigorous way. My background is in organic geochemistry and my comments focus mainly on the use of the alkenone $U^{k'}_{37}$ proxy in sediment and the overall structure and clarity of the paper. I cannot provide in-depth commentary on the quality/validity of the scanning methods used or the applied statistical analysis.

General Comments

1. The clarity of the manuscript could be improved in a few ways. First, I would recommend that the authors be much more specific about directionality when describing depth intervals, particularly given that this study explores both horizontal and vertical variability in $U^{k'}_{37}$. For example, I find the methods section beginning at line 69 quite confusing. A "30-cm long section" of the core is used, as well as three "5-cm replicated MSI measurements" of boxcore SPR0901-05BC. Later, "5-cm subsections" are "cut into 100-um slices" (line 78). I have a hard time visualizing this workflow, and I think that the schematic in Figure 1 could be introduced sooner, referenced more, and also made clearer. I don't quite understand what each step represents, particularly with respect to the direction of each core section and the direction of scanning. Arrows and labels would help.

2. The discussion of methods for spot-measuring alkenone compounds should be described more fully. The equation for $U^{k'}_{37}$ should be given somewhere, and I'm a bit confused about how swUk is calculated for the shot measurements. The authors say in Line 166 that "On average, at 62% of the shots at least one of the three compounds was detected...while at least one alkenone was detected in 46% and both in 34%." Were swUk calculated for shots with one alkenone compound present? If so, would these values not just go to 0 or 1? Is it that the

concentrations are near the detection limits of the instrument, or is it actually possible to have a relatively high concentration of C37:2, for example, and no C37:3? If the former is true, is there any size effect bias on swUk?

3. The paper needs the age controls on the sediment core to be discussed more thoroughly. The authors, for example, identify their sediments as containing the 1761 and 1532 CE gray flood layers (Line 115), but do not provide further information on how these specific floods were identified. A supplemental figure showing stratigraphic alignment of core KC1 with previously characterized SBB sediments, or a more thorough description of the method the authors used, would be helpful for demonstrating the validity of these flood ages to the reader. Further, the authors describe that SBB sediments are generally "varved, laminated, or bioturbated" (Line 112), but do not thoroughly describe the banding characteristics of the core section KC1. This creates confusion when the authors translate from depth to temporal scale.

In line 117, the authors begin by saying that only a linear interpolation age model is possible between 1761 and 1532, but later say that in core KC1, they detect SNRs "around 1 at interannual resolution, increasing to 3 at subdecadal resolution. The extracted noise component shows no dependence on time scale" (Line 315). The validity of this statement relies on there being an age model with annual to decadal resolution and an in-depth description of such age constraints. Ultimately, I'm confused as to how the authors translate between depth and age scales in core KC1.

4. How do $U^{k'}_{37}$ values derived from MSI scanning methods compare with measurements from traditional laboratory extractions and GC-MSD/GC-FID/HPLC measurements? I think that this technique is new enough that it warrants more thorough discussion/citation of previous validation studies.

5. In line 301, the authors speculate as to the causes of spatial heterogeneity in alkenone swUk and posit that this signal may be reflecting real variability (as opposed to measurement error), given that "water column and sediment trap data display a broader range of $U^{k'}_{37}$ values than core tops, and lab cultures have been shown to exhibit even greater variability..." I would appreciate a quantitative comparison between the variability captured by the MSI scanning techniques and those in the aforementioned settings. Is the degree of heterogeneity captured by MSI physically reasonable? I would particularly appreciate a direct comparison to sediment trap

data, given that the authors are investigating individual laminae which would accrete on somewhat comparable timescales. Based on previous studies, what is the maximum amount of noise that might reasonably be explained by variability in alkenone production about the same SST? What might be the approximate range of SST values that might be reasonably integrated over a single season? How do you expect water column and sedimentation processes to smooth (or not smooth) variability of alkenone production in the photic zone? It seems that if the observed spatial heterogeneity is *not* a product of measurement error, signal noise may contain valuable information that would be of interest to paleoclimatologists looking to apply MSI. I think that the paper could be improved if these possibilities were explored more thoroughly. I would recommend that Supplementary Figure 5 be brought into the main body of the paper and discussed.

**Corrections**

The manuscript contains a number of grammatical and typing errors and should be proofread, paying particular attention to comma and hyphen usage, spelling, and the use of "that" versus "which." Examples are presented for the first 100 lines and the Figure 2 caption, along with one citation recommendation.

*L30* **-** I suggest the authors use "longer" timescales rather than "slower"

*L31* - There is an extra space between in "dynamics of"

*L31* - Grammatical error in "Similarly, understanding of the long-term dynamics **of phenomena like monsoon** or El Nino-Southern Oscillation (ENSO), results…"

*L38* - Hyphen needed for "long-enough**"**

*L40* - Needs an additional comma. "This might, however, not be the case…"

L42 - Extra space between carriers and the following period

*L43-44* - Needs additional comma. "Such heterogeneity may arise during signal production, for example, due to…"

*L46* - I would drop the apostrophes/quotations for 'stratigraphic noise'

*L46* - Correct to "archives"

*L52* - Should cite Brassell et al., 1986

*L56* - Remove comma in "annual varve couplets, and"

*L57-59* - "for their preservation under low-oxygen conditions that reduce bioturbation" should be amended to "for their preservation under low-oxygen conditions, which reduce bioturbation" unless the authors mean to say that some low-oxygen conditions do not reduce bioturbation.

Similarly, "from minimal disturbances that preserved varves…" should be clarified.

*L64* - Need a comma before "which"... "individual MSI-based reconstructions, which indicate…"

*L69-70* - Opening line of the paragraph is not a full sentence and needs to be clarified: "Sediment core MV1012-001KC was retrieved by research vessel Melville during cruise MV1012 in 2012, stored and accessed at Scripps Institution of Oceanography's cored sediment and microfossil collection."

*L70-71* - Needs an additional hyphen for "30-cm-long". Comma usage with "named "KC1"" is incorrect. Need to change wording for the second clause of the sentence ending in "...in the following." For example: "...referred to as KC1 for the remainder of this manuscript."

*L72* - Correct to "three 5-cm-replicated measurements" for clarity

L72 - Need commas for "of boxcore SPR0901-05BC, originally published by Alfken et al. (2020), as an example…"

*L76* - Correct to "freeze-dried"

*L76* - Correct spelling of "embedded"

*L79* - Need a comma for "(Medite Cryostat M630), and"

*L81* - "Finally, the slices were measured"... what was measured exactly?

*L81* - Need a comma for "FT-ICR-MS, coupled to"

*L84* - Need to adjust "in the supplements section S1"... e.g. "in section S1 of the supplemental materials."

*L84* - "For each 5 cm depth"... In some places you use a dash with 5-cm, and other times you do not. Stay consistent with one or the other; both are correct.

*L88* – "adducts of the di- and triunsaturated alkenones" should be adjusted to "adducts of the di- and tri-unsaturated alkenones."

*L93* - Need a hyphen for "first-order estimation"

*L184* - Need to describe the Xs in panel A in the figure caption

*L184-185* - This part of the figure caption is not grammatically clear: "Maps are shown as measured, on the MSI coordinate grid, before affine transformation onto Xray coordinates, values below 1% and above 99% quantiles were removed for optimal color scaling during plotting.

---

## Author Comment (AC1)

**AC1: Response to reviewers' comments**

Colors:

Reviewer comments

Author response

Changed manuscript

**Response to Reviewer #1: RC1**
**(Comment on egusphere-2025-5089) by Joseph B. Novak**

Reviewer #1

SUMMARY

Viola et al. present a very nice study that explores the potential of mass spectrometry imaging to capture interannual climate variability in sedimentary archives. The data presented here probe the extent to which sedimentary processes may obscure these high-frequency climate signals. This study is an important contribution to the cutting edge of our field – particularly because of the profoundly important climate processes the MSI technique could be used to investigate. The writing and figure design are excellent. My main comment relates to the mass spectrometry techniques used. I am not familiar with the statistical techniques used in this manuscript and leave their evaluation to the other reviewers. I look forward to the publication of this work after my comments are addressed.

We thank the reviewer for helpful and constructive comments. As detailed below, we now have expanded the method section with further details on the analytical aspects of MSI based Uk'37 and their comparison to other approaches.

Reviewer #1:

MAJOR COMMENTS

Methods:

Is there any reason for concern about potential differences in the ionization efficiency of the C37:3 vs. C37:2 alkenones influencing your results? This is an issue for other mass spectrometer techniques (e.g., GC-MS and HPLC-MS, see (Chaler et al., 2000, 2003; Liao et al., 2023)). If steps were taken to account for this on the analytical side, some additional text outlining that procedure would be helpful to better understand the data.

Thank you for raising this point, as differences in the ionization efficiency between the alkenones would in fact have the potential to bias MSI derived Uk'37 series and reconstructed temperatures. This comment is related to general comments #2 of reviewer #2 and we changed the section according to both comments.

As the reviewer pointed out, issues regarding the ionization efficiencies of C37 alkenones are reported for several mass spectrometry approaches (Chaler et al., 2000, 2003; Liao et al., 2023). Unfortunately, to our knowledge, systematic studies addressing this for (MA)LDI FT-ICR MS workflows are currently lacking. Studies in warm SST areas (Napier et al., 2022; Obreht et al., 2022a; Wörmer et al., 2022) reported systematic offsets between GC-FID and MSI results and introduced site specific correction factors. Most likely these effects are due to a bias towards individual measurement spots ("shots") representing relatively colder temperatures. Shots with very high temperatures would lead to C37:3 more likely falling below the detection limit, and hence being excluded. In general, the potential influence of data processing choices is part of an ongoing project by the author. An alternative explanation could be related to lower alkenone concentrations at these areas in an effect similar to the "injection amount-effect" described in Liao et al. (Liao et al., 2023), especially given the still open questions around ion formation and matrix effects in (MA)LDI (Fuchs et al., 2010; Knochenmuss, 2006).

In the temperate Santa Barbara Basin, none of these biases would in principle play a big role, as both alkenone species are similarly abundant and concentrations are generally high. In fact, the comparisons to GC-FID data in sediments from the SBB showed high agreement between MSI and GC-FID based Uk'37 values (Alfken et al., 2020) (see Fig. R1 below). The MSI results for SBB sediments were additionally verified with instrumental SST data. The temporal overlap of boxcore SPR0901-05BC utilized by Alfken et al. 2020 and CalCOFI buoy station data (California State Department of Fish and Game; NOAA Fisheries; Scripps Institution of Oceanography, 2001) allowed to estimate the correlation between sediment MSI uk'37 based temperature reconstructions and buoy SST data of the modern era. On interannual timescale estimated from 1984 to 2009 the MSI based temperatures resulted in spearman's rank correlations of up to ~0.6 with upper 0-30m water column data at the closest CalCOFI station 81.8 46.9. Similar average temperatures for this period (MSI ~0.3C warmer) lead us to believe that differences in ionization efficiencies are not influential or relevant compared to the overall uncertainties and that MSI based uk37 in the middle of the Uk'37 temperature range and especially in SBB to be robust recorders of SST variability.

[Figure]

Fig. R1: Comparison of MSI-based and conventional extraction-based data of Uk'37. Grey points and line are MSI derived Uk'37 estimates, purple squares are GC-FID based estimates. Dotted rectangles indicate the corresponding depth range averaged for one sample used for extraction and horizontal lines display the corresponding mean value of the MSI data. Based on fig. 2 of Alfken et al. (2020).

The section now reads as:

*During the development of the MSI workflow by Wörmer et al, (Wörmer et al., 2014) and in subsequent studies, the resulting $U_{37}^{K'}$ values were verified using GC-FID measurements* (Alfken et al., 2020; Napier et al., 2022; Obreht et al., 2022a, b). *Differences in $U_{37}^{K'}$ values obtained by different mass spectrometry techniques have been reported and attributed to varying ionization efficiencies of alkenones, the co-elution of other compounds, and additional factors (Chaler et al., 2000, 2003; Liao et al., 2023; Rama-Corredor et al., 2018). Notably, data derived from MSI in warm SST regions have shown a cold bias compared to GC-FID data, leading to the introduction of site-specific correction factors. However, in the temperate SST regime of SBB, MSI and GC-FID data did not exhibit significant differences and correlated well with CalCOFI buoy SST data at the site (Alfken et al., 2020; California State Department of Fish and Game; NOAA Fisheries; Scripps Institution of Oceanography, 2001).*

Reviewer #1:

MINOR COMMENTS

L29–32: Another important aspect of paleoclimate archives is that they allow climate scientists to understand climate processes in warmer-than-present climate states. It may be worthwhile to add a comment to this effect.

Thank you, this aspect improves the rationale behind our study and was added accordingly.

The section now reads as:
*Understanding past climate and its dynamics is crucial for contextualizing recent climate change and processes in warmer-than-present climate states under projected future conditions.*

Reviewer #1:

L40: a comma is needed after "however."
L42: remove space between "carries" and the period at the end of the sentence.
L55: "export productivity" is a more appropriate term here than "primary productivity" because the sediments only preserve the proportion of the biological products that are exported from the surface and buried.
L150–151: I think there is a typo or missing word here.

Thank you for catching these errors and the suggestion, which clarifies what we are reconstructing with the pyropheophorbide α data.

The changes have been made accordingly, L150-151 now read as:
*Given a regional cluster of n proxy records with a similar climate between sites, the mean power spectrum, M, averaged across all individual records' spectra, will yield a precise estimate of the proxy spectrum P.*

Reviewer #1:

Figure 2 caption: an explanation of the black "Xs" in panel A would be nice.

Thank you for this suggestion as it makes the figure much easier to understand, especially when viewed in isolation.

The caption now reads as:
*Figure 2: Maps and time-series of the sediment section KC1. (A) X-ray density map with black crosses or rectangles where material was removed as markers for orientation of the samples, (B) exemplary swUk (spotwise $U_{37}^{K'}$) maps of one replicate per depth interval. Maps are shown as measured, on the MSI coordinate grid, before affine transformation onto Xray coordinates, values below 1% and above 99% quantiles were removed for optimal color scaling during plotting. (C) Scaled density plots of the spatial swUk maps per replicate, and (D) the resulting $U_{37}^{K'}$ time-series per replicate and depth interval. Note that the area corresponding to the 1761 AD floodlayer in depth interval 0-5 cm was removed prior to analyses.*

Reviewer #1:

L220: need a subscript on "C0"
L230: C37:3 needs a subscript

Thank you for catching these.

The lines have been changed accordingly.

**References**

Alfken, S., Wörmer, L., Lipp, J. S., Wendt, J., Schimmelmann, A., and Hinrichs, K.: Mechanistic Insights Into Molecular Proxies Through Comparison of Subannually Resolved Sedimentary Records With Instrumental Water Column Data in the Santa Barbara Basin, Southern California, Paleoceanography and Paleoclimatology, 35, https://doi.org/10.1029/2020PA004076, 2020.

California State Department of Fish and Game; NOAA Fisheries; Scripps Institution of Oceanography: Chemical, physical, and other data collected in the coastal waters of California as part of the California Cooperative Fisheries Investigation (CalCOFI) project since 1949, 2001.

Chaler, R., Grimalt, J. O., Pelejero, C., and Calvo, E.: Sensitivity Effects in Uk'37 Paleotemperature Estimation by Chemical Ionization Mass Spectrometry, Anal. Chem., 72, 5892–5897, https://doi.org/10.1021/ac001014q, 2000.

Chaler, R., Villanueva, J., and Grimalt, J. O.: Non-linear effects in the determination of paleotemperature *U*k'37 alkenone ratios by chemical ionization mass spectrometry, Journal of Chromatography A, 1012, 87–93, https://doi.org/10.1016/S0021-9673(03)01188-9, 2003.

Fuchs, B., Süss, R., and Schiller, J.: An update of MALDI-TOF mass spectrometry in lipid research, Prog Lipid Res, 49, 450–475, https://doi.org/10.1016/j.plipres.2010.07.001, 2010.

Knochenmuss, R.: Ion formation mechanisms in UV-MALDI, Analyst, 131, 966–986, https://doi.org/10.1039/B605646F, 2006.

Liao, S., Liu, X.-L., Manz, K. E., Pennell, K. D., Novak, J., Santos, E., and Huang, Y.: Comprehensive analysis of alkenones by reversed-phase HPLC-MS with unprecedented selectivity, linearity and sensitivity, Talanta, 260, 124653, https://doi.org/10.1016/j.talanta.2023.124653, 2023.

Napier, T. J., Wörmer, L., Wendt, J., Lückge, A., Rohlfs, N., and Hinrichs, K.-U.: Sub-Annual to Interannual Arabian Sea Upwelling, Sea Surface Temperature, and Indian Monsoon Rainfall Reconstructed Using Congruent Micrometer-Scale Climate Proxies, Paleoceanography and Paleoclimatology, 37, e2021PA004355, https://doi.org/10.1029/2021PA004355, 2022.

Obreht, I., De Vleeschouwer, D., Wörmer, L., Kucera, M., Varma, D., Prange, M., Laepple, T., Wendt, J., Nandini-Weiss, S. D., Schulz, H., and Hinrichs, K.-U.: Last Interglacial decadal sea surface temperature variability in the eastern Mediterranean, Nat. Geosci., 15, 812–818, https://doi.org/10.1038/s41561-022-01016-y, 2022a.

Obreht, I., De Vleeschouwer, D., Wörmer, L., Kucera, M., Varma, D., Prange, M., Laepple, T., Wendt, J., Nandini-Weiss, S. D., Schulz, H., and Hinrichs, K.-U.: Last Interglacial decadal sea surface temperature variability in the eastern Mediterranean, Nat. Geosci., 15, 812–818, https://doi.org/10.1038/s41561-022-01016-y, 2022b.

Rama-Corredor, O., Cortina, A., Martrat, B., Lopez, J. F., and Grimalt, J. O.: Removal of bias in C37 alkenone-based sea surface temperature measurements by high-performance liquid chromatography

fractionation, Journal of Chromatography A, 1567, 90–98, https://doi.org/10.1016/j.chroma.2018.07.004, 2018.

Wörmer, L., Elvert, M., Fuchser, J., Lipp, J. S., Buttigieg, P. L., Zabel, M., and Hinrichs, K.-U.: Ultra-high-resolution paleoenvironmental records via direct laser-based analysis of lipid biomarkers in sediment core samples, Proceedings of the National Academy of Sciences, 111, 15669–15674, https://doi.org/10.1073/pnas.1405237111, 2014.

Wörmer, L., Wendt, J., Boehman, B., Haug, G. H., and Hinrichs, K.-U.: Deglacial increase of seasonal temperature variability in the tropical ocean, Nature, 612, 88–91, https://doi.org/10.1038/s41586-022-05350-4, 2022.